# Drug Repositioning Applied to Cardiovascular Disease in Mucopolysaccharidosis

**DOI:** 10.3390/life12122085

**Published:** 2022-12-12

**Authors:** Gerda Cristal Villalba Silva, Thiago Steindorff, Roselena Silvestri Schuh, Natalia Cardoso Flores, Ursula Matte

**Affiliations:** 1Bioinformatics Core, Hospital de Clinicas de Porto Alegre, Ramiro Barcelos, Porto Alegre 2350, RS, Brazil; 2Gene Therapy Center, Hospital de Clinicas de Porto Alegre, Ramiro Barcelos, Porto Alegre 2350, RS, Brazil; 3Biomedical Sciences School, Institute of Health Sciences, UFRGS, Ramiro Barcelos, Porto Alegre 2600, RS, Brazil; 4Pharmaceutical Sciences Graduate Program, UFRGS, Avenida Ipiranga, Porto Alegre 2752, RS, Brazil; 5Genetics and Molecular Biology Graduate Program, UFRGS, Av. Bento Gonçalves, Porto Alegre 9500, RS, Brazil

**Keywords:** drug repositioning, lysosomal storage diseases, bioinformatics, gene expression analysis, cardiovascular diseases, systems pharmacology

## Abstract

Mucopolysaccharidoses (MPS) are genetic metabolic diseases characterized by defects in the activity of lysosomal hydrolases. In MPS, secondary cell disturbance affects pathways related to cardiovascular disorders. Hence, the study aimed to identify MPS-related drugs targeting cardiovascular disease and select a list of drugs for repositioning. We obtained a list of differentially expressed genes and pathways. To identify drug perturbation-driven gene expression and drug pathways interactions, we used the CMAP and LINCS databases. For molecular docking, we used the DockThor web server. Our results suggest that pirfenidone and colchicine are promising drugs to treat cardiovascular disease in MPS patients. We also provide a brief description of good practices for the repositioning analysis. Furthermore, the list of drugs and related MPS-enriched genes could be helpful to new treatments and considered for pathophysiological studies.

## 1. Introduction

Mucopolysaccharidoses (MPS) are a group of rare genetic metabolic diseases characterized by defects in the activity of lysosomal hydrolases that degrade glycosaminoglycans (GAGs), such as heparan and dermatan sulfate. Patients with MPS present a wide range of symptoms, characterizing a disease with progressive multisystemic involvement. Lysosomes are essential mediators of various cell processes, so multiple pathways are deranged in MPS patients. In addition, immune function may be impaired and directly impact disease pathogenesis (Bouhamdani et al., 2021) [1], with a particular focus on the Toll-like receptor 4 pathway (TLR4), as highlighted by Parker and Bigger (2019) [2].

Cardiovascular involvement is among the main features of MPS disorders, being a significant cause of morbidity and mortality. The range of manifestations includes progressive thickening and compromised function of the heart valves, conduction abnormalities, left ventricular hypertrophy, and diffuse coronary artery stenosis [3] (BRAUNLIN et al., 2011). The accumulation of GAGs within cardiovascular structures may likely originate these symptoms, leading to cellular dysfunction and widespread inflammation.

Repositioning analysis targeting lysosomal storage diseases, such as MPS, is gaining momentum. According to Schuchman and collaborators [4] (2021), no treatments are currently approved for nearly two-thirds of all lysosomal diseases. The opportunity to apply existing molecules to new indications was discussed by pharmaceutical companies (Cha et al., 2018) [5]. Nosengo and colleagues investigated the development costs for new therapeutic agents based on publicly available data (Wouters et al., 2020) [6]. There are several recent examples of repositioning drugs tested in vitro and patient-derived cells for lysosomal diseases, such as cystinosis (Bellomo et al., 2021) [7], Batten disease (Soldati et al., 2021) [8], Fabry (Garbade et al., 2020; Monticelli et al., 2022) [9,10], Gaucher (Pantoom et al., 2022) [11], and Niemann–Pick type C disease (Fukaura et al., 2021; Pepponi et al., 2022) [12,13]. Due to cost-effectiveness and a reduced timeline, repositioning drugs for new indications is a promising approach to finding disease treatments with compelling advantages over traditional drug development (Roessler et al., 2021) [14]. For this purpose, the main goal of this work is to provide a framework for drug repositioning analysis based on the transcriptional signatures of cardiac disease in MPS and to give an example with a case study to understand how to apply this approach to rare diseases.

## 2. Materials and Methods

### 2.1. Gene Expression Analysis

Transcriptome datasets were retrieved from the Gene Expression Omnibus database [15] (GEO, https://www.ncbi.nlm.nih.gov/geo/) (accessed on 5 October 2022) with the accession number GSE30657 (MPS VII, Mus musculus). We performed the gene expression analysis with the R package limma v.3.44.3 (Ritchie et al., 2015) [16]. More information about the datasets may be found in our database for differentially expressed genes in mucopolysaccharidoses (Soares et al., 2021) [17], MPSBase (https://www.ufrgs.br/mpsbase/) (accessed on 5 October 2022), and in Table 1. Transcriptome data was analyzed as differentially expressed genes (DEG), filtered by the false discovery rate (FDR) adjusted method. 

### 2.2. Gene Pathway Enrichment Analysis

We performed the pathway analysis using the R package pathfindR (Ulgen et al., 2019) [18], identifying active subnetworks and then performing enrichment analysis. Statistical analysis was performed by assessing the *p*-value using a hypergeometric distribution. Multiple test correction was also implemented by applying the FDR algorithm at a significance level of *p* < 0.05. Gene set annotations were obtained from the Kyoto Encyclopedia of Genes and Genomes database.

### 2.3. Drug Gene and Drug Pathway Interactions

To identify drug perturbation-driven gene expression and drug pathways interactions, we used the CMAP database (Lamb et al., 2006) [19] and CLUE platform (https://clue.io/about) (accessed on 5 October 2022), a cloud-based software platform for the analysis of perturbational datasets generated using gene expression (L1000) and proteomic (P100 and GCP) assays. In the CLUE platform, we used the repurposing app Touchstone to analyze datasets of chemical compounds and genetic perturbagenes where there are clinical studies that have reported gene expression signatures in cell lines. The Touchstone data provide a benchmark to assess connectivity among drugs and genes. To examine pathway drug interactions, we used the repurposing app to identify the drugs related to the pathways found in our enrichment results. The repurposing app provides information about 5000 compounds and drugs for drug discovery. We also used the Library of Integrated Network-based Cellular Signatures (LINCS, https://maayanlab.cloud/sigcom-lincs/) (accessed on 5 October 2022) to look for compounds that could modulate the activity of DEG in MPS.

### 2.4. Molecular Docking

The protein model was constructed using the SWISS-MODEL homology modeling web server, with the respective amino acid sequence as reference (Waterhouse et al., 2018) [20]. PDB ID was 2NRY and the template’s ID was 5uiu.2. The structural quality of the homology model was GMQE 0.91 and QMEANDDisCo Global was 0.88 ± 0.05. The mutated residue was altered to correspond to the wild-type sequence. The protonation assignment was performed using PROPKA (Olsson et al., 2011) [21]. The protonation states of the active site residues were also visually inspected. 

To assess the homology model structure’s quality, we have performed a Ramachandran plot with the Ramachandran plot server tool (Anderson et al., 2004) [22]. According to the plot, 95.865% of residues are observed in highly preferred conformations, 3.008% are in preferred conformations, and only 1.128% of residues showed questionable conformations (Appendix A). We’ve also performed the Verify3D analysis at UCLA’s Structure Validation Server (Bowie et al., 1991) [23]. According to it, 82.37% of the residues have averaged 3D-1D score greater or equal to 0.2. The high-quality cutoff percentage is considered to be 80% of residues. Both results attest that the homology model generated in SWISS-MODEL was adequate for the experiments.

To understand the molecular basis of the interactions between proteins and ligands, we used the DockThor web server v.2 (Santos et al., 2020; Guedes et al., 2021) [24,25]. The DockThor program is suitable for docking highly flexible and challenging ligands and is freely available as a virtual screening web server at https://www.dockthor.lncc.br/ (accessed on 5 October 2022). We used the first-rank RMSD, the program HADDOCK, and the Van der Waals energy scores to evaluate the best approach.

## 3. Results

### 3.1. Gene Expression Analysis

Drug repositioning starts with the identification of therapeutic targets. In the case of rare complex diseases, using high-throughput expression data may provide new targets. Here, a dataset from the MPS VII mouse model aorta (GSE30657) was used for two comparisons: MPS VII dilated aorta vs. WT control and MPS VII dilated aorta vs. MPS VII non-dilated aorta. The gene expression analysis results are summarized in Table 1, with 3973 and 1037 DEG, respectively. The complete gene expression table is available in Appendix A. 

Next, we searched which pathways were involved using the KEGG database [26] (Kanehisa et al., 2020). We found 222 KEGG pathways in the MPS VII dilated aorta vs. WT control comparison, out of which 116 were also present in the 117 pathways found in the MPS VII dilated vs. MPS VII non-dilated aorta comparison. Figure 1 demonstrates the top 20 KEGG pathways for the MPS VII dilated aorta vs. WT control (1A) and MPS VII dilated vs. MPS VII non-dilated aorta (1B). The most frequent pathways are related to immune system pathways, like the MAPK signaling pathway (with 70 differentially expressed genes), Th1 and Th2 cell differentiation (51 related genes), chemokine signaling pathway (47 genes), and NOD-like receptor signaling pathway (42 genes). In addition, we also identified several inflammasome-related genes, such as CASP1, CASP12, CASP5, CASP8, IRAK4, ITPR3, NLRP3, and TRAF. The complete KEGG pathway list is available in Appendix A. We also evaluated the interactions between the genes in each pathway, and the KEGG map for each pathway was constructed. An example of these analyses is shown in Figure 2. Overall results of gene expression and biological pathways show genes related to platelet-derived growth factor receptors (PDGFR), fibroblast growth factor receptor (FGFR), vascular endothelial growth factor receptor (VEGFR), epidermal growth factor receptor (EGFR), and rearranged during transfection (RET). 

### 3.2. Gene-Drug and Pathway-Drug Results

The 116 common pathways obtained above were used to identify gene-drug interactions. In total, 188 genes interact with one or more drugs. The most frequent drug types are kinase inhibitors, acting against JAK, PI3K, and MAPK. The complete list of genes and their target drugs is available in Appendix A. 

For the pathway-drug interactions, we searched for drugs acting as pathway inhibitors or inducers in the 116 common pathways shared between the comparisons, MPS VII dilated aorta vs. WT control, and MPS VII dilated aorta vs. MPS VII non-dilated aorta. In this step, we obtained 201 different drugs (Appendix A). 

### 3.3. Case Study: Molecular Docking of the IRAK4 and the Target Drugs

As immune-related pathways and inflammasome involvement were identified, we chose the IRAK4 (interleukin-1 receptor-associated kinase 4) protein for molecular docking due to its biochemical properties [27,28] (Janssens & Beyaert, 2003; Wang et al., 2009). According to the literature, we identified three main compounds targeting the IRAK4 enzyme (Manning et al., 2002) [29]. To assess their inhibition mechanisms, we performed the molecular docking of three related intracellular tyrosine kinase inhibitors: nintedanib, vandetanib, and gefitinib (Figure 3). The obtained affinity scores were −9.042 kcal/mol, −9.490 kcal/mol, and −9.987 kcal/mol. These binding energies were favorable for efficient docking, as Santos et al. (2020) [25] suggested. All of them had similar conformations inside the active site pocket and similar interactions with IRAK4′s residues. Gefitinib showed a probable hydrogen bond with Arg 273 main chain and a possible halogen bond between Lys 213 side chain and its chlorine atom. Nintedanib showed probable hydrogen bonding with the Lys 213 side chain’s nitrogen and the Met 265 main chain’s nitrogen. Arg 273 is also in H-bond proximity with the drug. Finally, vandetanib also had atoms in hydrogen bond proximity with the Arg 273 residue.

In order to validate the docking score results, we replicated the docking workflow with six other IRAK4 inhibitors with reported Ki or IC50 values. The results are shown in Table 2. Due to the small number of inhibitors found (3 with Ki and 3 with IC50 values), we were not able to calculate a correlation coefficient for docking scores and these values. Nonetheless, the affinity scores of the proposed inhibitors are similar, with mean = −9.105 ± 0.29. According to our model, gefitinib was the inhibitor with the highest affinity score (−9.987 kcal/mol) and ND-2110, the lowest (−8.334 kcal/mol). This lower affinity score is expected since it has the highest reported Ki value (7.5 nM). On the other hand, the molecule with the highest IC50, zabedosertib (3.4 nM), was the one with the second highest docking score (−9.546 kcal/mol), performing even better than vandetanib and nintedanib. Observing the molecular interactions, all four of them seem to be able to inhibit the enzyme through the same mechanism (hydrogen bonding with ARG 273). ND-2158 and ND-2110, which showed the lowest docking scores, form hydrogen bonds with ASP 272 rather than ARG 273. Overall, our data show that the affinity scores of gefitinib, vandetanib, and nintedanib are in order with the affinity scores of other reported IRAK-4 inhibitors.

## 4. Discussion

MPS patients suffer from different cardiac problems, mainly fibrosis of the conduction system with GAG infiltration (Braunlin et al., 2011) [3], valve stenosis (Boffi et al., 2018) [34], and left ventricular hypertrophy that contributes to morbidity and mortality. Current therapeutic options, such as hematopoietic stem cell transplantation (HSCT) and enzyme replacement therapy (ERT), may alter overall cardiovascular disease progression in MPS patients. However, specific tissues remain resistant to treatment and continue to manifest GAG storage (Poswar et al., 2022) [35]. Consequently, potentially life-threatening disease complications, such as those involving the cardiovascular system, remain untreated. Therefore, drug repositioning may offer a glimpse into novel adjuvant therapies for these diseases.

GAG overaccumulation has been associated with the release of various pro-inflammatory immune mediators and autophagy dysfunction. Furthermore, oxidative stress, abnormal mitochondrial function, disruption in ion homeostasis, and overexpression of lysosomal, and proteasomal-related genes also play a role in MPS disease (Fecarotta et al., 2020) [36]. In addition, chronic inflammatory diseases can be amplified by the activation of TLRs (Knowlton, 2017) [37]. Indeed, significant overexpression of the Toll-like receptor 4 (*TLR4*) gene, the activation of interleukin-1 receptor-associated kinase 4 (*IRAK4*), in addition to numerous cathepsin proteases and matrix metalloproteinases were observed in MPS cardiovascular disease (Stepien et al., 2020) [38]. 

One study suggested that heparan sulfate–TLR4-mediated monocyte/macrophage-induced inflammation contributes to the pathogenesis of cardiovascular disease in MPS I (Khalid et al., 2016) [39]. Furthermore, they reported alterations in gene and protein expression in arteries of the canine MPS I model, thus supporting the hypothesis that a GAG-induced inflammatory process is responsible for the pathogenesis of MPS I cardiovascular disease (Khalid et al., 2016) [39]. Another study demonstrated the involvement of the Toll-like receptor 4 pathway and the beneficial use of the TNF-α antagonist infliximab for the treatment of joint inflammation in MPS VI (Simonaro et al., 2010) [40].

From the results obtained by the drug-pathway interactions and the IRAK4 molecular docking, we showed some possible candidate drugs for MPS disorders with cardiovascular involvement, such as nintedanib, vandetanib, and gefitinib. They are potent and selective intracellular inhibitors of PDGFR, FGFR, VEGFR, EGFR, and RET tyrosine kinases. These small molecules have shown consistent antifibrotic, antiangiogenic, and anti-inflammatory activity in animal models (Wollin et al., 2015) [41]. However, with numerous clinical trials and a heavy focus on drug safety, many small-molecule kinase inhibitors (SMKIs) induce critical adverse events, such as cardiotoxicity (Jin et al., 2020) [42]. The primary cardiac adverse events were QT prolongation, hypertension, left ventricular dysfunction, arrhythmia, heart failure, and ischemia or myocardial infarction (Jin et al., 2020) [42]. Left ventricular dysfunction may be caused by nintedanib (Ameri et al., 2021) [43], and vandetanib can prolong the Q–T interval (Ton et al., 2013) [44]. Gefitinib may induce cardiotoxicity by modulating the cardiac pathways’ expression and function and forming CYP1A1-mediated reactive metabolites (Alhoshani et al., 2020; Korashy et al., 2016) [45,46]. It is essential to highlight that the reported cardiotoxicity with these agents is often but not always reversible (Zaborowska-Szmit et al, 2020; Upretty & Mansfield, 2020) [47,48]. New kinase inhibitors are designed to circumvent these adverse effects (Cohen et al., 2018; Lee et al., 2018) [49,50].

Other pathways enriched in the MPS VII aorta involved cell apoptosis, and several caspase genes were DEG. Gradual cardiomyocyte apoptosis occurs in failing hearts, leading to the progressive loss of cardiomyocytes and lethal heart failure (Gao et al., 2020) [51]. Angiotensin-converting enzyme inhibitors and β-receptor blockers are used in clinical practice, and these treatments’ benefits result partly from reductions in cardiomyocyte apoptosis. Moreover, impaired autophagy has been indicated to play a pathological role in the progression of heart failure. In this sense, another immunosuppressant drug candidate for MPS treatment could be sirolimus, an mTOR inhibitor that prevents apoptosis and promotes autophagy. Datasets comparisons in our study showed mTOR perturbations in cardiac disease caused by MPS; therefore, sirolimus could be used. This drug is often used as an immunosuppressant in cases of heart surgeries and the introduction of coronary stents (Gao et al., 2020; Shibata et al., 2019) [51,52]. However, potent immunosuppressants may pose some risks to MPS patients, as they may downregulate patients’ defenses from exogenous pathogens (Simonaro, 2016) [53].

Defective lysosomes lead to abnormal autophagy, inflammation, and reduced infection control, as occurs in MPS disorders (Simonaro, 2016) [53]. The inhibitory effect of autophagy on inflammasome activation in healthy cells has an important and broad impact. Defective autophagy leads to the accumulation of mitochondria in cells and elevated release of inflammasome activators through the production of reactive oxidative species (ROS); as such, autophagy may be required to remove aggregated inflammasome structures, thereby reducing pro-inflammatory responses (Huber & Teis, 2016; Maltez & Miao, 2016) [54,55]. Thus, lysosomes play a diverse and essential role in immunity and inflammation, partly through the regulation of autophagy, control of inflammasome release of cytokines, and the regulation of sphingolipid metabolism (Simonaro, 2016) [53]. Colchicine is an established anti-inflammatory drug, which attenuates NLRP3 (nucleotide-binding oligomerization domain-, leucine-rich repeat-, and pyrin domain-containing protein 3) inflammasome–mediated crystal-induced inflammation present in gout attributable to uric acid crystals and atherosclerosis attributable to cholesterol crystals (Martínez et al., 2018; Opstal et al., 2020) [56,57]. Although generally well tolerated at prescribed doses, colchicine has a narrow therapeutic window, with reported fatalities in patients with chronic renal insufficiency (Slobodnick et al., 2018) [58]. Moreover, colchicine neuromyopathy may occur with regular daily use, which is particularly undesirable in MPS patients, as they already suffer from neuropathies (Congedi et al., 2018) [59]. However, it could be a drug candidate in controlled settings and has never been tested for MPS.

MPS patients suffer from fibrosis of the conduction system with GAG infiltration (Braulin et al., 2011) [3], valve leaflets show prominent nodular fibrosis and calcification (Sherwood et al., 2021) [60], and end-stage disease is characterized by valve stenosis and fibrotic endocardium (Boffi et al., 2018) [34]. In addition, myocardial fibrosis can occur as a compensatory mechanism to replace cardiomyocyte necrosis and preserve the structural integrity of the myocardium. Nevertheless, progressive deposition of the extracellular matrix (ECM) due to a persistent injury may trigger a vicious cycle leading to persistent structural and functional alterations of the myocardium (Ma et al., 2018; Murtha et al., 2017) [61,62]. Although some drugs (e.g., inhibitors of the renin/angiotensin/aldosterone system, such as losartan) have been shown to reduce ECM deposition, no primarily antifibrotic medications are used to treat patients with MPS. Antifibrotic agents such as pirfenidone might inhibit fibroblast proliferation and collagen synthesis by interfering with transforming growth factor-β (TGF-β) and other fibrogenic growth factors, such as platelet-derived growth factor (PDGF) and basic fibroblast growth factor (bFGF). Pirfenidone also upregulates several matrix metalloproteinases (MMPs), attenuating ECM accumulation, and downregulates pro-inflammatory cytokines (such as tumor necrosis factor-α (TNF-α), interleukin (IL)-4, and IL-13), as well as inhibiting the formation of the NLRP3 inflammasome, which could modulate the inflammatory response and inhibit collagen synthesis (Aimo et al., 2022; Shah et al., 2021) [63,64]. Azambuja et al. (2021) [65] identified the activation of inflammasome-related proteins in the brain of MPS II mice and hypothesized that activation of the inflammasome cascade is related to neurological impairment. In this sense, there is a growing interest in investigating the role of pirfenidone in heart disease. As predicted by our computational study, MPS cardiac disease could also be a potential target.

For a successful targeted cardiovascular delivery of therapeutic agents, several barriers must be overcome, including the anatomical difficulties of access, the mechanical force of the blood flow, and the endothelial and cellular barriers [66] (Li et al., 2021). There are some strategies for overcoming these barriers, such as cardiac-specific and directed molecules or vectors, surgery, catheter-based delivery, intramyocardial injection, agents that increase vascular permeability (VEGF or nitroglycerin), bioengineered and cardiotropic vectors, and extracellular vesicles (Kulkarni et al., 2020) [67]; as well, new alternatives are constantly being researched (Sahoo et al., 2021) [68]. In addition, systemic administration is much less invasive and universally applicable but can cause off-target uptake in other organs and systems, and engineered drug-delivery systems may often be used, such as nanoparticles (Fan et al., 2020; Wang et al., 2021) [69,70], antibodies (Holland, 2017) [71], or ultrasound-targeted approaches (Holland, 2017) [71]. Injectable hydrogels, cellular and acellular material-based scaffolds, and nanoparticles are currently being investigated for the cardiac delivery of therapeutic agents (Pan et al., 2021) [72]. However, questions remain about the biocompatibility, targeting efficiency, immunogenicity, pro-inflammatory effects, degradation rates, clearance, and medical safety of these materials, which need to be carefully evaluated before developing clinical applications. Moreover, there is a possibility of using transcytotic mechanisms to reengineer biologics, such as using a mAb against an endogenous cardiac receptor transporter, which is expected to serve as a molecular shuttle to enhance the delivery of drugs to the heart (Sato et al., 2022) [73].

## 5. Conclusions

As for other orphan diseases, the need for more efficient adjuvant therapies for MPS can benefit from the pipeline used in this study, shown in Figure 4. Of course, one could analyze transcriptional regulators. Still, in this case, we wanted to present a more straightforward approach using differentially expressed genes, which are more widely available than a network of RNA regulators. In the present case, innate immunity and fibrosis appear to have a significant role in disease pathogenesis, as seen by the enriched DEG and pathway analysis. From the gene-drug and pathway-drug interaction lists, we chose a few candidate drugs, such as pirfenidone and colchicine, that could have a synergistic benefit in MPS patients with heart disease. It must be noted that these drugs would not be curative but may improve quality of life and slow disease progression, reducing patient morbidity. These drugs’ clinical benefits and safety must be addressed in animal models before clinical studies can be performed on their efficacy in MPS patients. Nevertheless, the approach shown here may speed the discovery of new therapeutic targets with low cost and high specificity.

## Figures and Tables

**Figure 1 life-12-02085-f001:**
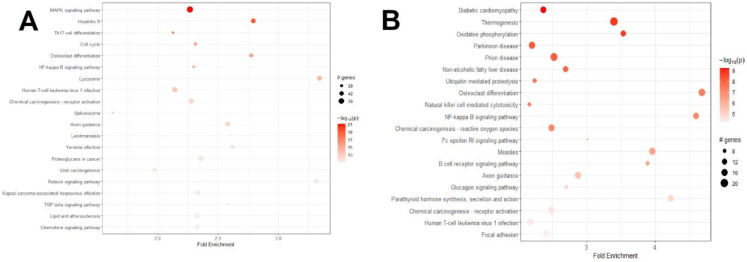
Top 20 KEGG pathways for the differentially expressed genes. The circles are related to the number of enriched genes in each pathway. The red scale indicates the *p*-value. (**A**): MPS VII dilated aorta vs. WT control enriched pathways. (**B**): MPS VII dilated vs. MPS VII non-dilated aorta enriched pathways.

**Figure 2 life-12-02085-f002:**
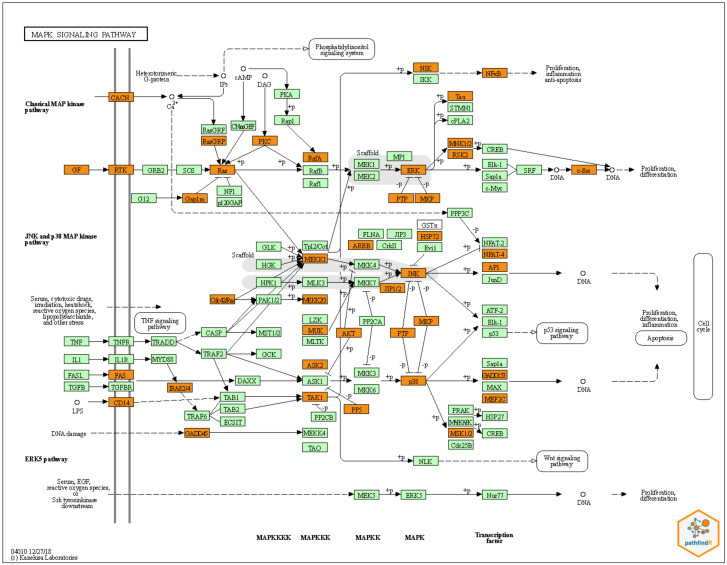
KEGG map and KEGG-related genes interaction network. To characterize the analysis, we choose the pathway with the most abundant differentially expressed genes: the MAPK signaling pathway. Genes in orange are present in our list; mint green represents the other genes in the pathway.

**Figure 3 life-12-02085-f003:**
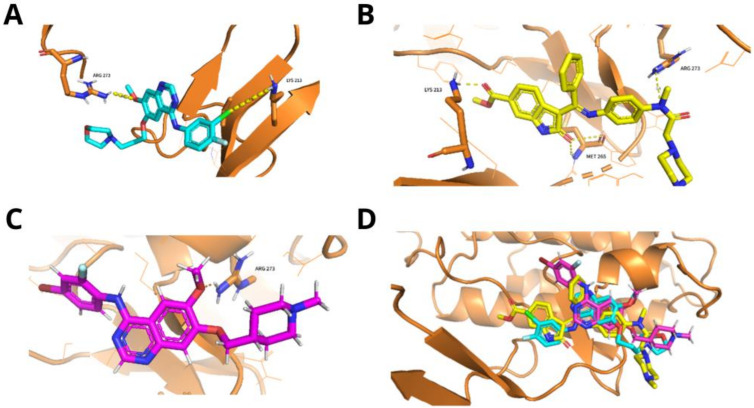
Molecular docking results for IRAK4 enzyme and their target drugs. (**A**) Gefitinib (PubChem: 123631, affinity score: −9.042). (**B**) Nintedanib (PubChem: 135423438, affinity score: −9.490). (**C**) Vendetanib (PubChem: 3081361, affinity score: −9.987). (**D**) Superposing of IRAK4 and all three target drugs. All of them had similar conformations inside the active site pocket and similar interactions with IRAK4′s residues. Blue: gefitinib; yellow: nintedanib; magenta: vandetanib.

**Figure 4 life-12-02085-f004:**
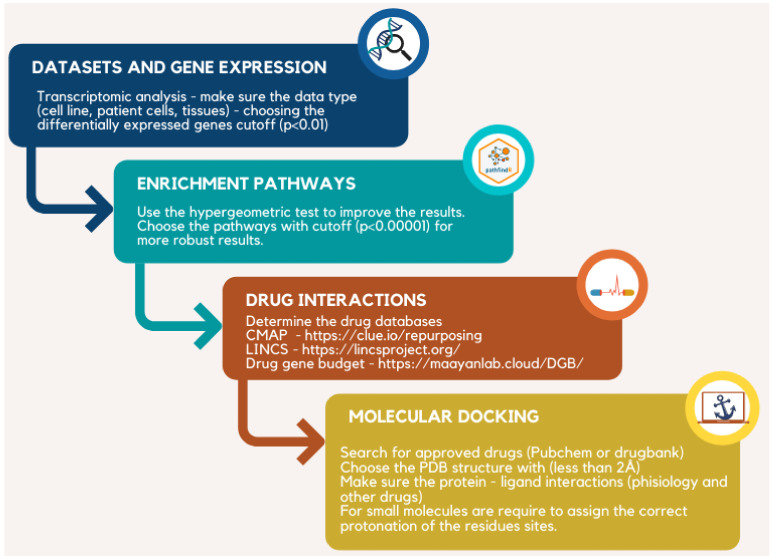
Good practices to perform repositioning analysis.

**Table 1 life-12-02085-t001:** Number of differentially expressed genes (DEG) up and down regulated for each comparison.

Comparison	DEGs	UP	DOWN
MPS VII Dilated aorta vs. WT control	3973	2296	1677
MPS VII Dilated aorta vs. MPS VII non dilated aorta	1037	444	593

**Table 2 life-12-02085-t002:** Affinity, IC50 and Ki for different IRAK4 inhibitors.

Inhibitor	Affinity	IC50	Ki	References
Gefitinib	−9.987	NA	NA	
Zabedosertib	−9.546	3.4 nM	NA	[30]
Vandetanib	−9.49	NA	NA	
Compound 1	−9.318	NA	1.2 nM	[28]
Nintedanib	−9.042	NA	NA	
Rac-45	−8.996	1 nM	NA	[31]
Zimlovisertib	−8.771	2 nM	NA	[32]
ND-2158	−8.461	NA	1.3 nM	[32,33]
ND-2110	−8.334	NA	7.5 nM	[33]

## Data Availability

The data are available in the GEO NCBI with the accession number GSE30657. The gene expression analysis is found at https://www.ufrgs.br/mpsbase/. All the code and data are available at https://github.com/Kur1sutaru/drugrepositioningcardiomps.

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
