# Peer review of "Drug Repositioning Applied to Cardiovascular Disease in Mucopolysaccharidosis"

_life, 2022, doi:10.3390/life12122085_

Round 1
Reviewer 1 Report
The paper : Drug repositioning applied to cardiovascular disease in Muco- 2 polysaccharidosis by Ursula Matte and co-workers should be published but it needs major revisions.
In the first place, transcriptomic analyses have already been carried out by several groups, Wgrzyn and the same Matte for example.
I think their results should be compared to those in the present paper.
FOr example:
Bobrowski L, “Łukaszuk T, Gaffke L, et al. Separating gene clustering in the rare mucopolysaccharidosis disease [published correction appears in J Appl Genet. 2022 Sep;63(3):595]. J Appl Genet. 2022;63(2):361-368. doi:10.1007/s13353-022-00691-2
Gaffke L, Pierzynowska K, Krzelowska K, Piotrowska E, Wgrzyn G. Changes in expressions of genes involved in the regulation of cellular processes in mucopolysaccharidoses as assessed by fibroblast culture-based transcriptomic analyses. Metab Brain Dis. 2020;35(8):1353-1360. doi:10.1007/s11011-020-00614-2
Rintz E, Gaffke L, Podlacha M, et al. Transcriptomic Changes Related to Cellular Processes with Particular Emphasis on Cell Activation in Lysosomal Storage Diseases from the Group of Mucopolysaccharidoses. Int J Mol Sci. 2020;21(9):3194. Published 2020 Apr 30. doi:10.3390/ijms21093194
Network Analysis Reveals Proteins Associated with Aortic Dilatation in Mucopolysaccharidoses
Thiago Corrêa, Bruno César Feltes, Esteban Alberto Gonzalez, Guilherme Baldo & Ursula Matte
The authors state There are a few examples of repositioning drugs tested in vitro and in patient-derived cells for lysosomal diseases (Garbade et al., 2020; Bellomo et al., 2021; Soldati et al., 2021)” this is not true
By searching PubMed with the words “drug repositioning lysosomal disease” several results are obtained hence the word “ few”is misleading. At least the most recent papers should cited besides those chosen by the authors:
Monticelli M, Liguori L, Allocca M, et al. Drug Repositioning for Fabry Disease: Acetylsalicylic Acid Potentiates the Stabilization of Lysosomal Alpha-Galactosidase by Pharmacological Chaperones. Int J Mol Sci. 2022;23(9):5105. Published 2022 May 4. doi:10.3390/ijms23095105
Pantoom S, Hules L, Schöll C, et al. Mechanistic Insight into the Mode of Action of Acid “-Glucosidase Enhancer Ambroxol. Int J Mol Sci. 2022;23(7):3536. Published 2022 Mar 24. doi:10.3390/ijms23073536
Pepponi R, De Simone R, De Nuccio C, et al. Repurposing Dipyridamole in Niemann Pick Type C Disease: A Proof of Concept Study. Int J Mol Sci. 2022;23(7):3456. Published 2022 Mar 22. doi:10.3390/ijms23073456
Kopytova AE, Rychkov GN, Nikolaev MA, et al. Ambroxol increases glucocerebrosidase (GCase) activity and restores GCase translocation in primary patient-derived macrophages in Gaucher disease and Parkinsonism. Parkinsonism Relat Disord. 2021;84:112-121. doi:10.1016/j.parkreldis.2021.02.003
The authors used a transcriptomic analysis to identify target proteins and in silico docking to find drugs that binds the targets. This protocol will not lead to specific drugs that bind the protein that is responsible for the disease. Nonetheless I believe it can be a useful approach but other approaches to reposition drugs for rare diseases have been proposed and should be cited:
Cha, Y.; Erez, T.; Reynolds, I.J.; Kumar, D.; Ross, J.; Koytiger, G.; Kusko, R.; Zeskind, B.; Risso, S.; Kagan, E.; et al. Drug Repurposing from the Perspective of Pharmaceutical Companies. Br. J. Pharmacol. 2018, 175.
Nosengo, N.; Wouters, O.J.; McKee, M.; Luyten, J. Estimated Research and Development
Investment Needed to Bring a New Medicine to Market, 2009-2018. JAMA - J. Am. Med. Assoc. 2020, 323.
Hay Mele B, Citro V, Andreotti G, Cubellis MV. Drug repositioning can accelerate discovery of pharmacological chaperones. Orphanet J Rare Dis. 2015;10:55
Brasil S, Allocca M, Magrinho SCM, et al. Systematic Review: Drug Repositioning for Congenital Disorders of Glycosylation (CDG). Int J Mol Sci. 2022;23(15):8725. Published 2022 Aug 5. doi:10.3390/ijms23158725
The authors chose to approximate the variance to constant during the model-building procedure. However, the visualization of the mean-variance scatter plot suggests a consistent mean-variance trend. Why do the authors not calculate precision weights for that by incorporating a call to voom() into their limma code
In M&M, the authors state they use two GEO studies (GSE78889, GSE30657). There is also an explanation of the procedure to convert dog and mouse genes to human gene ID. However, in the rest of the paper, the authors only refer to GSE30657. Thus, the authors should remove GSE78889 from M&M.
The authors should add more information on the Swiss-Model subsection of M&M, e.g., the ID of the sequence used for the model. I am also wondering how the homology model performs with respect to the alpha fold one (readily available and indexed in the ebi database).
The author dedicates an M&M subsection to ZINC, but they use it only for surveying drug status. In my opinion, they should either remove ZINC from M&M and cite it in the results. Alternatively, they could increase the information gathered from it.
I have a problem with references, the authors should use a consistent style for all of them.

Author Response
In the first place, transcriptomic analyses have already been carried out by several groups, Węgrzyn and the same Matte for example. I think their results should be compared to those in the present paper.
Indeed the mentioned works deal with transcriptomic analyses but our focus was on using transcriptomic data for drug repositioning.
The authors state “There are a few examples of repositioning drugs tested in vitro and in patient-derived cells for lysosomal diseases (Garbade et al., 2020; Bellomo et al., 2021; Soldati et al., 2021)” this is not true
By searching PubMed with the words “drug repositioning lysosomal disease” several results are obtained hence the word “ few”is misleading. At least the most recent papers should cited besides those chosen by the authors:
Monticelli M, Liguori L, Allocca M, et al. Drug Repositioning for Fabry Disease: Acetylsalicylic Acid Potentiates the Stabilization of Lysosomal Alpha-Galactosidase by Pharmacological Chaperones. Int J Mol Sci. 2022;23(9):5105. Published 2022 May 4. doi:10.3390/ijms23095105
Pantoom S, Hules L, Schöll C, et al. Mechanistic Insight into the Mode of Action of Acid β-Glucosidase Enhancer Ambroxol. Int J Mol Sci. 2022;23(7):3536. Published 2022 Mar 24. doi:10.3390/ijms23073536
Pepponi R, De Simone R, De Nuccio C, et al. Repurposing Dipyridamole in Niemann Pick Type C Disease: A Proof of Concept Study. Int J Mol Sci. 2022;23(7):3456. Published 2022 Mar 22. doi:10.3390/ijms23073456
Thank you very much for pointing this mistake. We fixed the text and we added the suggested references.
The authors used a transcriptomic analysis to identify target proteins and in silico docking to find drugs that binds the targets. This protocol will not lead to specific drugs that bind the protein that is responsible for the disease. Nonetheless I believe it can be a useful approach but other approaches to reposition drugs for rare diseases have been proposed and should be cited:
Cha, Y.; Erez, T.; Reynolds, I.J.; Kumar, D.; Ross, J.; Koytiger, G.; Kusko, R.; Zeskind, B.; Risso, S.; Kagan, E.; et al. Drug Repurposing from the Perspective of Pharmaceutical Companies. Br. J. Pharmacol. 2018, 175
Nosengo, N.;Wouters, O.J.; McKee, M.; Luyten, J. Estimated Research and Development Investment Needed to Bring a New Medicine to Market, 2009-2018. JAMA - J. Am. Med. Assoc. 2020, 323.
Hay Mele B, Citro V, Andreotti G, Cubellis MV. Drug repositioning can accelerate discovery of pharmacological chaperones. Orphanet J Rare Dis. 2015;10:55
The authors chose to approximate the variance to constant during the model-building procedure. However, the visualization of the mean-variance scatter plot suggests a consistent mean-variance trend. Why do the authors not calculate precision weights for that by incorporating a call to voom() into their limma code
We believe using the voom function was unnecessary regarding our data type, since our data was mainly from microarray studies. We understand that the mean-variance trend is converted by the voom function into precision weights, which are incorporated into the analysis of log-transformed RNA-seq counts using the same linear modeling commands as microarrays.
In M&M, the authors state they use two GEO studies (GSE78889, GSE30657). There is also an explanation of the procedure to convert dog and mouse genes to human gene ID. However, in the rest of the paper, the authors only refer to GSE30657. Thus, the authors should remove GSE78889 from M&M.
Thank you very much for the suggestion. We removed the GSE78889 from the methods section.
The authors should add more information on the Swiss-Model subsection of M&M, e.g., the ID of the sequence used for the model. I am also wondering how the homology model performs with respect to the alpha fold one (readily available and indexed in the ebi database).
Thank you for your comment, we added more information in the methods section. Unfortunately, we did not had time to perform the analysis with alpha fold. If more time is given, these data could be included.
The author dedicates an M&M subsection to ZINC, but they use it only for surveying drug status. In my opinion, they should either remove ZINC from M&M and cite it in the results. Alternatively, they could increase the information gathered from it.
We removed ZINC from the methods section.
I have a problem with references, the authors should use a consistent style for all of them.
Thank you for the suggestion. We revised all the reference formats.

Reviewer 2 Report
Thank you for giving me the opportunity to review the manuscript “Drug repositioning applied to cardiovascular disease in Mucopolysaccharidosis” by Cristal Villalba Silva et al.
The article describes a possible way to identify drugs for the successful treatment of disease that have been approved or are being tested in other pathologies in order to obtain accelerated approval. The approach is called drug repositioning. This approach is very interesting from the point of view of this reviewer and especially for the rare diseases such as lysosomal storage diseases of which the mucopolysaccharidoses (especially type VII) are concerned here. Nevertheless, I see room for some improvements in the manuscript. I will try to make my suggestions intelligible here.
Major:
- There are definitely different views on what drug repositioning exactly means and some authors make a distinction between drug repurposing and drug repositioning. The terms seem to be used synonymously in the manuscript. Since these terms are central to the article, I suggest that the authors choose a more concrete description or definition of both terms here to help the reader.
- In general, the workflow is not completely clear to me. Intuitively, it is clear, but why do you first look for potential targets in transcriptomics and then look for direct interactors (inhibitors and inducers)? Is it not (logically) conceivable to look for transcriptional regulators instead of physical interactors? This could be taken up once for the sake of logical derivation. Maybe you can argue that this approach would be too unspecific or is it maybe too difficult to look for RNA transcriptional regulators? In case you can provide a specific desired transcriptional signature you can then identify interesting hit compounds via LINCS or cmap?
- Although the methods section 2.1 mentions working with "GSE78889 (MPS I, Canis lupus)", the presented datasets are all based on MPSVII as far as I can tell. Thus, this passage could be removed. In general, the applicability of the results to MPS-I and II can perhaps be discussed rather than making it a fixed part of the study.
- I understand the authors wanted to limit the molecular docking analysis to one example, but is there a rational to choose IRAK4? The authors write that it is an inflammasome-related gene along with CASP1, CASP12, CASP5, CASP8, ITPR3, NLRP3, and TRAF. It is not part of the MAPK pathway as shown in Figure 2. Wouldn't it be more logical to pick a gene here?
Minor:
- The authors write, "There are a few examples of repositioning drugs tested in vitro and in patient-derived cells for lysosomal diseases (Garbade et al., 2020; Bellomo et al., 2021; Soldati et al., 2021)." I would beg to differ. The references here seem somewhat aimless, which may seem to support the statement, but review articles and one on ceroid lipofuscinosis are used. In my view, as presented, it hardly represents the state of the art in research. The Ambroxol/Gaucher topic alone is a typical repositioning example. Cyclodextrin and Niemann-Pick type C and Bortezomib and Pompe come to mind off the top of my head. I think there are many examples of this, not only sporadically. On the contrary, almost the entire community is currently trying to reposition known drugs.
- A color code for the mint green and orange genes should be provided in Figure 2.
- There is a mix-up of the suppl. tables. There are 4 files, but only three are referenced in the text. In the supplement part at the end of the script even only two of them.
- Spelling and grammar. Although I am generally not qualified to make higher claims about language use, I do notice some inconsistencies. The words enzyme and physiology or sometimes spelled incorrectly “enzime” (Figure legend 3) and “phisiology” (Figure 4). In the same figure 4 I would replace “make sure the data type” and “make sure protein - ligand interactions” with “validate data type” etc. IRAK4 is sometimes written like this and sometimes with a dash (IRAK-4). In addition, the abbreviation (interleukin-1 receptor-associated kinase 4) is explained later in the text instead of the first time it is mentioned.
Author Response
- There are definitely different views on what drug repositioning exactly means and some authors make a distinction between drug repurposing and drug repositioning. The terms seem to be used synonymously in the manuscript. Since these terms are central to the article, I suggest that the authors choose a more concrete description or definition of both terms here to help the reader.
Thank you for your comment. Indeed, most authors use both terms interchangeably. According to your suggestion we standardized the use of drug repositioning.
- In general, the workflow is not completely clear to me. Intuitively, it is clear, but why do you first look for potential targets in transcriptomics and then look for direct interactors (inhibitors and inducers)? Is it not (logically) conceivable to look for transcriptional regulators instead of physical interactors? This could be taken up once for the sake of logical derivation. Maybe you can argue that this approach would be too unspecific or is it maybe too difficult to look for RNA transcriptional regulators? In case you can provide a specific desired transcriptional signature you can then identify interesting hit compounds via LINCS or cmap?
Yes we could have analyzed transcription factors but we were interested into looking specifically to differentially expressed genes as part of the strategy (since data from differentially expressed genes is more widely available). We have added a comment on that at the conclusions section.
In general, the applicability of the results to MPS-I and II can perhaps be discussed rather than making it a fixed part of the study.
We have tried to show the applicability of the results to MPS I and II. Maybe you could be more specific on your suggestion?
- I understand the authors wanted to limit the molecular docking analysis to one example, but is there a rational to choose IRAK4? The authors write that it is an inflammasome-related gene along with CASP1, CASP12, CASP5, CASP8, ITPR3, NLRP3, and TRAF. It is not part of the MAPK pathway as shown in Figure 2. Wouldn't it be more logical to pick a gene here?
IRAK4 is shown in figure 2 (in the left hand side, at the bottom). Since the gene list does not present genes in any particular order, IRAK4 was picked as it is a reasonably well described enzyme and serves to show how docking can be used as the next step in drug repositioning.
Minor:
- The authors write, "There are a few examples of repositioning drugs tested in vitro and in patient-derived cells for lysosomal diseases (Garbade et al., 2020; Bellomo et al., 2021; Soldati et al., 2021)." I would beg to differ. The references here seem somewhat aimless, which may seem to support the statement, but review articles and one on ceroid lipofuscinosis are used. In my view, as presented, it hardly represents the state of the art in research. The Ambroxol/Gaucher topic alone is a typical repositioning example. Cyclodextrin and Niemann-Pick type C and Bortezomib and Pompe come to mind off the top of my head. I think there are many examples of this, not only sporadically. On the contrary, almost the entire community is currently trying to reposition known drugs.
Thank you for the suggestion, we have modified this topic in the manuscript. The following reference was also included:
Fukaura, M., Ishitsuka, Y., Shirakawa, S., Ushihama, N., Yamada, Y., Kondo, Y., Takeo, T., Nakagata, N., Motoyama, K., Higashi, T., Arima, H., Kurauchi, Y., Seki, T., Katsuki, H., Higaki, K., Matsuo, M., & Irie, T. (2021). Intracerebroventricular Treatment with 2-Hydroxypropyl-β-Cyclodextrin Decreased Cerebellar and Hepatic Glycoprotein Nonmetastatic Melanoma Protein B (GPNMB) Expression in Niemann-Pick Disease Type C Model Mice. International journal of molecular sciences, 22(1), 452. https://doi.org/10.3390/ijms22010452
- A color code for the mint green and orange genes should be provided in Figure 1.
We fixed the legend in figure 1.
- There is a mix-up of the suppl. tables. There are 4 files, but only three are referenced in the text. In the supplement part at the end of the script even only two of them.
Thank you for your remarks, we have fixed this issue.
- Spelling and grammar. Although I am generally not qualified to make higher claims about language use, I do notice some inconsistencies. The words enzyme and physiology or sometimes spelled incorrectly “enzime” (Figure legend 3) and “phisiology” (Figure 4). In the same figure 4 I would replace “make sure the data type” and “make sure protein - ligand interactions” with “validate data type” etc. IRAK4 is sometimes written like this and sometimes with a dash (IRAK-4). In addition, the abbreviation (interleukin-1 receptor-associated kinase 4) is explained later in the text instead of the first time it is mentioned.
Thank you for the suggestions.
Reviewer 3 Report
Silva et al. submitted title “Drug repositioning applied to cardiovascular disease in Mucopolysaccharidosis” to investigate the putative drug design and development where authors enclosed an in-silico study on the targets retrieved from their gene and pathway study.
Points that need to be addressed
1. Authors performed molecular docking of the IRAK4 and the target drugs, where no figures regarding molecular interactions were disclosed. Please incorporate figures showing the molecular interactions.
2. In section 2.5, the authors quoted homology modeling; however, no relevant information was provided in the following manuscript. If homology modeling was performed, specify the selected template, and structural quality of the homology model.
3. On page 4, lines 175, 176, and 177, the authors quoted, “These binding energies were favorable for efficient docking, as Santos et al. (2020) suggested. All of them had similar conformations inside the active site pocket and similar interactions with IRAK-4’s residues”
To justify this statement of similar conformation, the authors need to add the RMSD values and a figure showing a superpose of all three conformations.
4. Authors do consider docking scoring in terms of affinity; however, how was the docking score validated in the first place was not included in the manuscript. In my suggestion, authors could consider at least 4-5 known inhibitors with Ki values (even irrelevant to the current drug target as what is considered in their study) and then compare their docking scores with the reported Ki values, so that the limitation of the docking method can be evaluated. Please add relevant information.
The authors need to address the following points before consideration of the presented manuscript.
Author Response
- Authors performed molecular docking of the IRAK4 and the target drugs, where no figures regarding molecular interactions were disclosed. Please incorporate figures showing the molecular interactions.
Thank your suggestion, we have replaced the images in figure 3 to better show the molecular interaction.
- In section 2.5, the authors quoted homology modeling; however, no relevant information was provided in the following manuscript. If homology modeling was performed, specify the selected template, and structural quality of the homology model.
Thank you for pointing out. We have added such details.
- On page 4, lines 175, 176, and 177, the authors quoted, “These binding energies were favorable for efficient docking, as Santos et al. (2020) suggested. All of them had similar conformations inside the active site pocket and similar interactions with IRAK-4’s residues”
To justify this statement of similar conformation, the authors need to add the RMSD values and a figure showing a superpose of all three conformations.
We also added a figure showing the superposing of all three molecules interacting with IRAK4.
- Authors do consider docking scoring in terms of affinity; however, how was the docking score validated in the first place was not included in the manuscript. In my suggestion, authors could consider at least 4-5 known inhibitors with Ki values (even irrelevant to the current drug target as what is considered in their study) and then compare their docking scores with the reported Ki values, so that the limitation of the docking method can be evaluated. Please add relevant information.
Indeed this a relevant topic. In order to validate the docking score results we replicated the docking workflow with six other IRAK4 inhibitors with reported Ki or IC50 values for this enzyme. The results are shown in table 2. Our data shows that the affinity scores of geftnib, vandetanib and nintedanib are in order with the affinity scores of other reported IRAK-4 inhibitors.
Round 2
Reviewer 1 Report
The paper is now acceptable for publication
Author Response
Thank you.
Reviewer 2 Report
Dear authors,
thanks for re-submitting the above article. The manuscript has also clearly improved in light of the response to Reviewer #1 and #3's comments. I congratulate the authors for the successful revision. I still have two important points of correction.
1. In my first assessment I wrote: "In general, the applicability of the results to MPS-I and II can perhaps be discussed rather than making it a fixed part of the study."
The authors replied: "We have tried to show the applicability of the results to MPS I and II. Maybe you could be more specific on your suggestion?"
I would like to specify: The authors state: "This work provides an example of this application in lysosomal storage diseases, applied to cardiovascular impairment in Mucopolysaccharidosis type I and VII."
This is the second sentence of the Abstract. However, throughout the whole manuscript there is just one paragraph in the discussion mentioning MPS I and even less mentioning of MPS II. The data shown exclusively deal with MPS VII. I think MPS I should not be mentioned here so dominantly as if the manuscript showed data on it. I think this is misleading. That is my point. The manuscript deals with MPS VII - full stop.
2. Reviewer #1 and myself requested to better reflect the state-of-the-art in drug repositioning in LSD research. Reviewer #1 suggests to mention certain references (Monticelli et al. (2022), Pantoom et al. (2022) and Pepponi et al. (2022)). The authors implemented them in the reference list, but they are currently not mentioned in the text. It was mentioned to me that the reference Fukaura et al. (2021) was added in this context. This is indeed found in both places. I suggest that the other references mentioned should still be added in the text, as they are all very recent and reflect well the strong effort of drug repositioning in the LSD community.
Author Response
Thank you for your comments.
- We removed the reference to MPS I in the abstract. In the main text, references to MPS I or MPS II were maintained in the discussion section, as examples.
- Indeed we mentioned the works of Pepponi et al on Niemman-Pick type C and Pantoom et al on Gaucher but we forgot to add their names into the text. We have now corrected this mistake.
Reviewer 3 Report
The authors did revise most of the comments except comment 2, which was based on homology modeling. However, some basic information about homology was surely added on page 3. However, this is inconsistent with current standards.
To the author's attention, GMQE, and QMEANDDisCo Global value, which I believe authors retrieved from the swiss model, is generally insufficient markers to provide the structural quality of the built homology model. Therefore, other evaluations must be done.
Homology modeling is a tool that can translate the primary amino acid sequence into a three-dimensional structure. However, to ensure the quality of homology modeling- other important key parameters are conventionally incorporated, such as
Geometry of dihedral angles of the homology model (by using Ramachandran plot),
Energy model between template and homology model (such as ProSA), or
Verify3d graph between template and homology model.
for more information, authors could use Table 1 from the paper as a reference-10.1016/j.ejmech.2019.04.064
Author Response
We thank the reviewer for his/hers kind comments.
Regarding Ramachandran plot and Verify3D analysis: To assess the homology model structure’s quality we’ve performed Ramachandran plot with
Ramachandran plot server tool. According to the plot, 95.865% of residues are observed in highly preferred conformations, 3.008% are in preferred conformations and only 1.128% of residues showed questionable conformations. We’ve also performed the Verify3D analysis at UCLA’s Structure
Validation Server. According to it, 82.37% of the residues have averaged 3D-1D score greater or equal to 0.2. The high-quality cutoff percentage is considered to be 80% of residues. Both results show that the homology model generated in SwissModel was adequate for the experiments. These results were added to the text and Ramachandran's plot was included as a supplementary figure.
